# Management of Pregnancy in a Patient with Familial Hypercholesterolemia and Previous Myocardial Infarction—Treatment with LDL Apheresis: A Case Report

**DOI:** 10.3390/reports7020039

**Published:** 2024-05-19

**Authors:** Milos Milincic, Jovana Todorovic, Stefan Dugalic, Ivana Novakovic, Maja Macura, Katarina Lalic, Miroslava Gojnic Dugalic

**Affiliations:** 1Clinic for Gynecology and Obstetrics, University Clinical Centre of Serbia, 11000 Belgrade, Serbia; stef.dugalic@gmail.com (S.D.); ivananovakovic223@gmail.com (I.N.); maja_macura@live.com (M.M.); miroslavagojnicdugalic@yahoo.com (M.G.D.); 2Faculty of Medicine, University of Belgrade, 11000 Belgrade, Serbia; jole6989@hotmail.com (J.T.); katarina.s.lalic@gmail.com (K.L.); 3Institute of Social Medicine, University Clinical Centre of Serbia, 11000 Belgrade, Serbia; 4Clinic of Endocrinology, University Clinical Centre of Serbia, 11000 Belgrade, Serbia

**Keywords:** pregnancy, familial hypercholesterolemia, LDL apheresis, pregnancy outcomes

## Abstract

Familial hypercholesterolemia, a genetic disorder marked by elevated low-density lipoprotein cholesterol (LDL-C), poses significant risks for premature atherosclerosis and cardiovascular diseases, particularly during pregnancy. One of the safe methods of treating this condition in pregnancy is with the use of LDL apheresis. We present a 38-year-old primigravida with homozygous Familial Hypercholesterolemia (HoFH), ischemic cardiomyopathy, and angina pectoris. Two years before conception, extremely elevated lipid levels prompted statin therapy and lifestyle changes. Stent placements followed acute myocardial infarction. When planning pregnancy, statins were discontinued, but lipid levels elevated. LDL apheresis was initiated, achieving a 60% reduction. Throughout pregnancy, 16 LDL apheresis sessions were performed every 14 days, maintaining optimal lipid profiles. A cesarean section was performed in the 38th week of gestation, delivering a healthy infant. The patient resumed statin therapy after 8 months of breastfeeding. The patient maintained cardiovascular health, demonstrating the feasibility of controlled HoFH pregnancies. This case highlights the successful management of HoFH during pregnancy using LDL apheresis, ensuring maternal and fetal well-being. Future research on novel treatments and their safety during pregnancy is essential for refining therapeutic approaches in similar cases.

## 1. Introduction

Familial hypercholesterolemia (FH) is a genetic disorder characterized by elevated levels of low-density lipoprotein cholesterol (LDL-C) in the blood, leading to premature atherosclerosis and myocardial infarction (MI) [1]. Women with FH are at an increased risk of cardiovascular complications during pregnancy, like accelerated atherosclerosis, pre-eclampsia, venous thromboembolism, arrhythmias, worsening of valvular heart disease, etc. [2]. The physiological changes that occur during pregnancy, such as increased blood volume and altered lipid metabolism, can exacerbate the existing lipid abnormalities in FH. This increased risk necessitates careful monitoring of lipid levels and cardiovascular health throughout the pregnancy. Statins, fibrates, and other commonly used medications to lower cholesterol are generally contraindicated during pregnancy due to potential teratogenic effects and proven increases in abortion rates [3]. LDL apheresis is a method aimed at selectively removing LDL cholesterol from the bloodstream, thereby reducing the risk of cardiovascular events. Previous literature has underscored the efficacy of LDL apheresis in mitigating cardiovascular risk factors in patients with refractory hypercholesterolemia. For instance, a study by Thompson et al. demonstrated significant reductions in LDL cholesterol levels following LDL apheresis sessions, accompanied by improvements in endothelial function and arterial stiffness [4]. Similarly, the findings of the study conducted by Taylan et al. highlighted the favorable impact of LDL apheresis on lipid profiles and cardiovascular outcomes in patients with familial hypercholesterolemia [5]. In the context of pregnancy-related complications, limited data exist regarding the use of LDL apheresis. However, anecdotal evidence and case reports, such as the present one, suggest that LDL apheresis may be considered in pregnant women with severe hypercholesterolemia and a history of cardiovascular events. Notably, careful monitoring and individualized treatment strategies are essential to ensure maternal and fetal well-being throughout the pregnancy and postpartum period. This case report outlines the utilization of LDL apheresis therapy in managing a 38-year-old woman experiencing her first pregnancy, diagnosed with HoFH, and with a history of MI occurring during gestation and the subsequent period of breastfeeding.

## 2. Detailed Case Description

A 38-year-old primigravida came to our gynecology clinic at 10 weeks of spontaneous pregnancy with homozygous familial hypercholesterolemia (HoFH) and previously implanted coronary stents after acute myocardial infarction (MI).

Two years before pregnancy, the patient was examined by an endocrinologist for the first time due to an elevated lipid profile: total cholesterol (TC) 11.29 mmol/L, LDL-C 9.25 mmol/L, triglycerides (TG) to 2.35 mmol/L, and high-density lipoprotein cholesterol (HDL-C) 1.73 mmol/L. Genetic testing and family history revealed HoFH. Three months of statin therapy (Rosuvastatin tablets 40 mg daily in a single dose) and a cholesterol-lowering diet showed a decrease in TC to 5.52 mmol/L, LDL-C to 3.49 mmol/L, TG to 1.14 mmol/L, and HDL-C to 1.51 mmol/L. In the same year, due to anginal pains, a coronary angiography was performed, revealing narrowing of the right coronary artery (RCA) and left anterior descending artery (LAD). Two percutaneous coronary interventions were conducted, with two RCA stents and two LAD stents placed.

A year later, to conceive, statin therapy was discontinued. Subsequently, lipid levels rose, with TC 10.03 mmol/L and LDL-C 8.10 mmol/L. Considering high lipid values in the preconception period, therapy with LDL apheresis commenced for secondary prevention of cardiovascular events and stent restenosis. The cholesterol-lowering diet continued. The results of the first LDL apheresis were the following: TC 11.18 mmol/L to 4.71 mmol/L, HDL-C 1.39 mmol/L to 1.01 mmol/L, LDL-C 8.72 mmol/L to 2.87 mmol/L, and TG 2.35 mmol/L to 1.82 mmol/L. Six months after the first LDL apheresis, the patient came for the first clinical gynecological exam at 10 weeks of pregnancy. The pregnancy treatment plan was created in collaboration with an endocrinologist and a cardiologist. During pregnancy, serial LDL apheresis was performed at the clinic for endocrinology every 14 days. Direct adsorption of lipoproteins (DALI) was used as a method of apheresis that lasted approximately 3 h, treating a blood volume of 5000 mL, and proceeded without any subjective or objective complaints. A total of 16 apheresis courses and routine cardiologist check-ups involving echocardiography were carried out. Biochemical analyses conducted before and after apheresis revealed a substantial 60% reduction in TC/LDL-C. No cardiac therapy was administered during pregnancy, and no cardiac symptoms were recorded. The pregnancy was appropriately managed without complications. By cardiological recommendations, a cesarean section was conducted during the 38th week of gestation. A female infant was delivered, measuring 47 cm in length, weighing 2560 g, with a head circumference of 34 cm, and achieving an Apgar score of 9. The postpartum period proceeded without complications, and the patient continued LDL apheresis every 24–30 days throughout the 8-month breastfeeding period, after which the lipid-lowering therapy (Rosuvastatin tablets 40 mg daily in a single dose, and Ezetimib tablets 10 mg daily in a single dose) was resumed successfully.

## 3. Discussion

FH constitutes a group of genetic defects leading to profound elevations in blood cholesterol levels and a heightened risk of premature coronary heart disease (CHD) [6]. Previously, pregnancy with HoHF was thought to be very risky [7]. New research indicates that it is possible to control the disease during pregnancy and give birth to healthy fetuses [8,9]. The clinical diagnosis is made by a positive family history of early coronary disease and LDL-C levels over 4.9 mmol/L for heterozygous carriers and over 12.9 mmol/L for homozygous carriers [6,10]. The reasons for LDL apheresis therapy in the case of our patient and in pregnancy in general are as follows: statins have shown great effectiveness in lowering blood cholesterol. Still, they are contraindicated during pregnancy and lactation [11]. Other non-statin drugs, such as ezetimibe, fibrates, and niacin, are associated with teratogenic effects. PCSK9 inhibitors are a possible treatment in the future if they prove safe in pregnancy [11]. Bile acid sequestrants have minimal potential fetal risks. However, their use is constrained by side effects such as elevated TG levels and constipation [6]. In cases of extremely high LDL-C values during pregnancy, therapeutic LDL apheresis is indicated [12]. Our patient started statin therapy in her early thirties, shortly after an acute myocardial infarction (MI) and, consequently, four coronary stents. In the preconception period, statin therapy was discontinued, and regular LDL apheresis was performed up to 8 months postpartum. We were guided by the previously successful treatment of pregnant women with LDL apheresis [9].

Our patient gave birth at 38 weeks of pregnancy to a healthy female newborn weighing 2560 g, which is considered small for gestational age (SGA). Elevated LDL-C values can disrupt the circulation of the uteroplacental unit, slowing blood flow and increasing vascular resistance, which leads to fetal growth restriction [13]. Confounding factors in the case of our patient could be obesity, with her body mass index (BMI) of 27.2 kg/m^2^, which has been associated with an increased risk of having an SGA child. Furthermore, disturbance in lipoprotein metabolism leads to increased oxidation, free radicals, and damage to the vascular endothelium. All of these can lead to disorders in the spectrum of preeclampsia (PE) [14]. In our patient, arterial pressure values (between 100/60 mmHg and 120/80 mmHg), the presence of edema, and proteinuria were thoroughly monitored and in order. During pregnancy, the ultrasound stigmata of SGA and PE, as well as the flows through the umbilical and uterine arteries, were also monitored regularly and serially, and they were also within normal limits.

Individuals with FH inherit a mutated gene that impairs the body’s ability to effectively remove LDL-C from the bloodstream, resulting in prolonged exposure to high cholesterol levels. This prolonged exposure significantly contributes to the development of atherosclerosis and other cardiovascular issues [6]. Our goals were to maintain LDL-C values below 8 mmol/L, prevent further complications of the maternal cardiovascular system, and deliver a pregnancy with a healthy fetus at term. During pregnancy, LDL apheresis was performed every 14 days with the DALI system at the endocrinology clinic. LDL-C values before treatment were between 12 and 14 mmol/L, and after 3 h of treatment, between 5.5 and 7.8 mmol/L, which we considered very successful in comparison to the literature [8,15,16]. We report that the total time off statin therapy in our patient was 22 months, from the beginning of pregnancy planning to 8 months postpartum while she was breastfeeding. There were no cardiovascular complications during this period. Regular examinations by cardiologists, echocardiography, and Color-Doppler sonography of lower extremities were being implemented during pregnancy and in the postpartum period.

The importance of our work is reflected in the decision to initiate LDL apheresis therapy for secondary prevention of cardiovascular events and stent restenosis in the preconception period and throughout pregnancy, which represents a significant advancement in clinical management. This proactive approach underscores the importance of individualized treatment strategies to optimize maternal and fetal outcomes in high-risk pregnancies complicated by severe hypercholesterolemia and cardiovascular comorbidities. Additionally, the successful pregnancy outcome without cardiovascular complications, coupled with the absence of adverse effects on the newborn despite the prolonged period without statin therapy, highlights the efficacy and safety of LDL apheresis as a therapeutic option in pregnant women with severe hypercholesterolemia and cardiovascular disease history. This case underscores the potential of LDL apheresis to fill the therapeutic gap in cholesterol management during pregnancy, where traditional pharmacological interventions are limited by safety concerns.

Future research for FH treatment is necessary. The safety and efficacy of highly effective cholesterol-lowering PCSK9 inhibitors during pregnancy are limited, and further studies are needed. Mipomersen and Lomitapide are novel medications that may be considered in specific cases, but their safety during pregnancy is not well-established. It is important to note that any treatment decisions during pregnancy must weigh the potential benefits for the mother against the potential risks to the developing fetus.

## Data Availability

The original contributions presented in the study are included in the article, further inquiries can be directed to the corresponding author.

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
