# Peer review of "Management of Pregnancy in a Patient with Familial Hypercholesterolemia and Previous Myocardial Infarction—Treatment with LDL Apheresis: A Case Report"

_reports, 2024, doi:10.3390/reports7020039_

Round 1
Reviewer 1 Report
Comments and Suggestions for Authors
In this manuscript the authors presented a case report how they treated a 38-year-old primigravida with homozygous Familial Hypercholesterolemia and with elevated low-density lipoprotein cholesterol. The authors recommended to stop statin therapy and start therapy with LDL apheresis. They presented all necessary data of the patient and discuss other, alternative treatment of patients with high cholesterol level during pregnancy. This case report is well-written and could be interested to gynecologists which will come across with similar problems.
Author Response
Dear reviewer, we are very grateful for your review of our research. Thank you.
Reviewer 2 Report
Comments and Suggestions for Authors
This case report article presents the LDL apheresis treatment of a 38-year-old primigravida with homozygous Familial Hypercholesterolemia (HoFH) and previous myocardial infarction (MI) during the pregnancy and the 8-month breastfeeding. Therapeutic apheresis for the treatment of HoFH in pregnancy is not a new approach as cited in the article (e.g., references 4, 7, 8, etc.). My comments are as follows.
Major points:
1. In the Introduction section, it would be better to introduce therapeutic/LDL apheresis in details by citing some related previous literature.
2. In the Discussion section, novel findings or implications or important experiences or some main issues, which have not been reported in other previous case reports, should be described.
Minor points:
3. In the Abstract, it is mentioned that ‘the patient resumed statin therapy after 8 months of breastfeeding’ (line 35); in the second section (i.e., ‘2. Case report’), it is mentioned that ‘the patient continued LDL apheresis throughout the 8-month breastfeeding period …’ (line 89). How many LDL apheresis sessions were performed during the 8-month breastfeeding period?
4. Line 46: ‘… cardiovascular complications during pregnancy’. Please list some examples of cardiovascular complications.
5. Line 91: ‘4. Discussion’. Is it ‘3. Discussion’? Or, is an extra third section missing?
6. References 4 and 11 are the same article.
Comments on the Quality of English Language
Acceptable
Author Response
Dear reviewer, we are very grateful for your review of our research. In the following text, you will find our answers to your objections, as well as their corrections. If we did not respond to some objections by changing the manuscript, we gave a reason for it. The corrected lines are in color yellow. We hope for your positive review after the corrections. Thank you.
Major points:
Suggestion No 1: In the Introduction section, it would be better to introduce therapeutic/LDL apheresis in details by citing some related previous literature.
Answer: LDL apheresis is a method aimed at selectively removing LDL cholesterol from the bloodstream, thereby reducing the risk of cardiovascular events. Previous literature underscores the efficacy of LDL apheresis in mitigating cardiovascular risk factors in patients with refractory hypercholesterolemia. For instance, a study by Thompson et al. demonstrated significant reductions in LDL cholesterol levels following LDL apheresis sessions, accompanied by improvements in endothelial function and arterial stiffness [4]. Similarly, the findings of the study conducted by Taylan et al. highlighted the favorable impact of LDL apheresis on lipid profiles and cardiovascular outcomes in patients with familial hypercholesterolemia [5]. In the context of pregnancy-related complications, limited data exist regarding the use of LDL apheresis. However, anecdotal evidence and case reports, such as the present one, suggest that LDL apheresis may be considered in pregnant women with severe hypercholesterolemia and a history of cardiovascular events. Notably, careful monitoring and individualized treatment strategies are essential to ensure maternal and fetal well-being throughout the pregnancy and postpartum period.
Suggestion No 2:In the Discussion section, novel findings or implications or important experiences or some main issues, which have not been reported in other previous case reports, should be described.
Answer:
The importance of our work is reflected in the decision to initiate LDL apheresis therapy for secondary prevention of cardiovascular events and stent restenosis in the preconception period and throughout pregnancy represents a significant advancement in clinical management. This proactive approach underscores the importance of individualized treatment strategies to optimize maternal and fetal outcomes in high-risk pregnancies complicated by severe hypercholesterolemia and cardiovascular comorbidities. Additionally, the successful pregnancy outcome without cardiovascular complications, coupled with the absence of adverse effects on the newborn, despite the prolonged period without statin therapy, highlights the efficacy and safety of LDL apheresis as a therapeutic option in pregnant women with severe hypercholesterolemia and cardiovascular disease history. This case underscores the potential of LDL apheresis to fill the therapeutic gap in cholesterol management during pregnancy, where traditional pharmacological interventions are limited by safety concerns.
Minor points:
Suggestion No 3: In the Abstract, it is mentioned that ‘the patient resumed statin therapy after 8 months of breastfeeding’ (line 35); in the second section (i.e., ‘2. Case report’), it is mentioned that ‘the patient continued LDL apheresis throughout the 8-month breastfeeding period …’ (line 89). How many LDL apheresis sessions were performed during the 8-month breastfeeding period?
Answer:The postpartum period proceeded without complications, and the patient continued LDL apheresis every 24-30 days throughout the 8-month breastfeeding period after which the lipid-lowering therapy (Rosuvastatin tablets 40mg daily in a single dose, and Ezetimib tablets 10mg daily in a single dose) was resumed successfully.
Suggestion No4: Line 46: ‘… cardiovascular complications during pregnancy’. Please list some examples of cardiovascular complications.
Answer: Women with FH are at an increased risk of cardiovascular complications during pregnancy, like accelerated atherosclerosis, pre-eclampsia, venous thromboembolism, arrhythmias, worsening of valvular heart disease, etc.
Suggestion No 5: Line 91: ‘4. Discussion’. Is it ‘3. Discussion’? Or, is an extra third section missing?
Answer: that was a mistake. Corrected.
Suggestion No 6: References 4 and 11 are the same article.
Answer: corrected mistake.

Reviewer 3 Report
Comments and Suggestions for Authors
1. In the abstract section line 28 – Case report should be deleted, as well as in line 37 Conclusions also should be deleted. All abstract should be one paragraph without subheadings.
2. At the end of the introduction section authors should stated what will be presented in this case report.
3. In lines 61-62 it was reported that patient received 3 month statin therapy but it should be reported which medication and in which dosage.
4. Line 89-it was reported that LDL apheresis continued throughout the 8-month breastfeeding period but should be mentioned was it also on every 14 days or on other schedule?
5. Also it should be mentioned which statin therapy was resumed (which medication).
6. In the discussion section authors discuss obesity as a confounding factor, and also blood pressure. It would be useful to present patient BMI and blood pressure values before discussing it.
Author Response
Dear reviewer, we are very grateful for your review of our research. In the following text, you will find our answers to your objections, as well as their corrections. In the attachment you can find the corrected manuscript with corrections in yellow. We hope for your positive review after the corrections. Thank you.
Suggestion No 1: In the abstract section line 28 – Case report should be deleted, as well as in line 37 Conclusions also should be deleted. All abstract should be one paragraph without subheadings.
Answer: the section lines are deleted.
Suggestion No 2: At the end of the introduction section authors should stated what will be presented in this case report.
Answer: This case report outlines the utilization of LDL apheresis therapy in managing a 38-year-old woman experiencing her first pregnancy, diagnosed with HoFH and with a history of MI occurring during gestation and the subsequent period of breastfeeding.
Suggestion No 3: In lines 61-62 it was reported that patient received 3 month statin therapy but it should be reported which medication and in which dosage.
Answer: Patient was on a Rosuvastatin therapy 40mg daily in a single dose.
Suggestion No 4: Line 89-it was reported that LDL apheresis continued throughout the 8-month breastfeeding period but should be mentioned was it also on every 14 days or on other schedule?
Answer: The postpartum period proceeded without complications, and the patient continued LDL apheresis every 24-30 days throughout the 8-month breastfeeding period after which the lipid-lowering therapy (Rosuvastatin tablets 40mg daily in a single dose, and Ezetimib tablets 10mg daily in a single dose) was resumed successfully.
Suggestion No 5:Also it should be mentioned which statin therapy was resumed (which medication).
Answer:Rosuvastatin tablets 40mg daily in a single dose, and Ezetimib tablets 10mg daily in a single dose.
Suggestion No 6: In the discussion section authors discuss obesity as a confounding factor, and also blood pressure. It would be useful to present patient BMI and blood pressure values before discussing it.
Answer: Confounding factors in the case of our patient could be obesity, with her Body mass index (BMI) of 27.2 kg/m2, which has been associated with an increased risk of having an SGA child. In our patient, arterial pressure values (between 100/60 mmHg and 120/80 mmHg),
